# Functional Connectivity between the Resting-State Olfactory Network and the Hippocampus in Alzheimer’s Disease

**DOI:** 10.3390/brainsci9120338

**Published:** 2019-11-25

**Authors:** Jiaming Lu, Nicole Testa, Rebecca Jordan, Rommy Elyan, Sangam Kanekar, Jianli Wang, Paul Eslinger, Qing X. Yang, Bing Zhang, Prasanna R. Karunanayaka

**Affiliations:** 1Department of Radiology, The Pennsylvania State University College of Medicine, Hershey, PA 17033, USA; b101230039@gmail.com (J.L.); nikkites58@gmail.com (N.T.); rjordan@pennstatehealth.psu.edu (R.J.); relyan@pennstatehealth.psu.edu (R.E.); peslinger@pennstatehealth.psu.edu (P.E.); qyang@pennstatehealth.psu.edu (Q.X.Y.); 2Medical School of Nanjing University, Nanjing 210008, China; zhangbing_nanjing@vip.163.com; 3Department of Neurology, The Pennsylvania State University College of Medicine, Hershey, PA 17033, USA; 4Department of Neurosurgery, The Pennsylvania State University College of Medicine, Hershey, PA 17033, USA; 5Department of Neural and Behavioral Sciences, The Pennsylvania State University College of Medicine, Hershey, PA 17033, USA; 6Department of Public Health Sciences, The Pennsylvania State University College of Medicine, Hershey, PA 17033, USA

**Keywords:** olfactory network, hippocampus, functional connectivity, resting state, Alzheimer’s disease

## Abstract

Olfactory impairment is associated with prodromal Alzheimer’s disease (AD) and is a risk factor for the development of dementia. AD pathology is known to disrupt brain regions instrumental in olfactory information processing, such as the primary olfactory cortex (POC), the hippocampus, and other temporal lobe structures. This selective vulnerability suggests that the functional connectivity (FC) between the olfactory network (ON), consisting of the POC, insula and orbital frontal cortex (OFC) (Tobia et al., 2016), and the hippocampus may be impaired in early stage AD. Yet, the development trajectory of this potential FC impairment remains unclear. Here, we used resting-state functional magnetic resonance imaging (rs-fMRI) data from the Alzheimer’s Disease Neuroimaging Initiative (ADNI) to investigate FC changes between the ON and hippocampus in four groups: aged-matched cognitively normal (CN), early mild cognitive impairment (EMCI), late mild cognitive impairment (LMCI), and AD. FC was calculated using low frequency fMRI signal fluctuations in the ON and hippocampus (Tobia et al., 2016). We found that the FC between the ON and the right hippocampus became progressively disrupted across disease states, with significant differences between EMCI and LMCI groups. Additionally, there were no significant differences in gray matter hippocampal volumes between EMCI and LMCI groups. Lastly, the FC between the ON and hippocampus was significantly correlated with neuropsychological test scores, suggesting that it is related to cognition in a meaningful way. These findings provide the first in vivo evidence for the involvement of FC between the ON and hippocampus in AD pathology. Results suggest that functional connectivity (FC) between the olfactory network (ON) and hippocampus may be a sensitive marker for Alzheimer’s disease (AD) progression, preceding gray matter volume loss.

## 1. Introduction

Alzheimer’s disease (AD) is marked by the insidious onset of episodic memory loss, but early clinical symptoms also include sensory (i.e., vision, hearing, and olfaction) and motor impairments [1,2,3,4,5,6]. Of particular interest are the well-documented olfactory deficits that often precede cognitive impairment [1,7,8]. Moreover, these early-stage olfactory deficits often predict the onset of dementia [7,9,10,11,12,13,14]. A post-mortem study demonstrated a significant correlation between patients with impaired odor identification and increased density of tangles in the entorhinal cortex and CA1/subiculum region of the hippocampus [15]. These findings allude to widespread cortical neuronal loss that severely compromises anatomical and functional connections within and between sensory and cognitive networks in AD, namely olfaction and memory [1].

There is ample evidence suggesting that olfactory network (ON) impairment is a potential biomarker for detecting the initial neuropathological processes observed in AD [16,17,18]. Emerging evidence suggests that these olfactory impairments may predict the onset of AD, amnestic mild cognitive impairment (aMCI), and the presence of amyloid-β (Aβ) and tau pathology in cognitively normal adults. However, very few AD studies have investigated olfactory deficits from a brain network perspective. This is a critical step in understanding the relationship between AD pathology and olfactory deficits. More specifically, a network approach may clarify if olfactory deficits, such as odor-identification deficits, are to be linked to damage in the primary and secondary olfactory structures, or if they reflect secondary effects of damage in higher-order cortical areas affected by AD.

Studies of resting-state (RS) functional connectivity (FC) in AD holds great promise in revealing how brain network dynamics are altered by neurodegeneration [19]. Several resting-state functional magnetic resonance imaging (rs-fMRI) studies have reported characteristic disruptions in functional networks prominent in AD pathology [20,21,22]. Most RS brain networks are intrinsically organized, but there is evidence to suggest that these networks are also influenced by task performance [23,24]. Furthermore, while FC assesses the integration of spontaneous and evoked brain activity across distant brain regions and networks, it is also thought to be shaped by the brains’ structural connectivity [25,26]. The resting-state ON is functionally connected to the hippocampus, which anchors the episodic memory system (Tobia et al., 2016) [27]. Additionally, task fMRI studies have shown similar FC patterns with the hippocampus [28], a finding that is consistent with behavioral studies linking olfactory performance to memory function [10,11,29].

It must be noted that whether olfactory deficits are due to a dysfunction in the central or peripheral olfactory nervous system remains unclear. For example, using task fMRI, Vasavada et al. (2017) suggested that olfactory deficits are most likely caused by degeneration in the central olfactory nervous system. Likewise, Zhang et al. (2018) showed that the number of mature olfactory sensory neurons in the olfactory epithelium is reduced in apolipoprotein E (ApoE)-deficient mice and suggested a link between the olfactory mucosa (peripheral) and the pathogeneses of AD.

The goal of this study was to investigate the resting-state functional connectivity (RS-FC) between the ON and hippocampus using the Alzheimer’s Disease Neuroimaging Initiative (ADNI) data. We analyzed the ON FC in subjects who were categorized as age-matched cognitively normal (CN), early mild cognitive impairment (EMCI), late mild cognitive impairment (LMCI), and AD. In line with the disconnection hypothesis of AD, widespread neuronal loss and brain network dysfunction were expected to cause olfactory impairments. Since olfactory deficits emerge early in the pathological cascade, we hypothesized that loss of functional coherence in olfactory structures may disrupt ON FC patterns [30,31,32]. Further, given that cognitive decline becomes progressively worse across the disease state, we expected a similar trajectory of impairment in the FC between the ON and the hippocampus.

## 2. Materials and Methods

### 2.1. Participants

Resting-state fMRI data from 147 subjects (male, designated as “M”) were obtained from the ADNI (http://www.adni-info.org/) dataset. Of the 147 subjects, 44 subjects were designated as CN (mean age = 74.18, 17 M), 46 as EMCI (mean age = 71.69, 19 M), 31 as LMCI (mean age = 72.41, 18 M), and 26 as AD (mean age = 71.55, 11 M) during the initial visit of the ADNI-GO or ADNI-2 phases (Table 1). EMCI and LMCI subjects were diagnosed based on the criteria described in the ADNI-2 procedure manual (http://www.adni-info.org/). Briefly, the criteria for diagnosing a subject with E/LMCI were as follows: (1) a subjective memory concern as reported by the subject, study partner, or clinician; (2) abnormal memory function documented by performance on the Logical Memory II subscale (Delayed Paragraph Recall, Paragraph A only) from the Wechsler Memory Scale—Revised (the maximum score is 25) based on the following education adjusted cutoffs—[a] 9 to 11 for sixteen or more years of education, [b] 5 to 9 for 8 to fifteen years of education, and [c] 3 to 6 for zero to seven years of education—(3) the Mini-Mental State Exam (MMSE) score is between 24 and 30 (inclusive); (4) if subjects with less than 8 years of education score outside of this inclusion range, exceptions may be made at the discretion of the project director; (5) Clinical Dementia Rating score of 0.5, with a Memory Box score of at least 0.5; and (6) general cognition and functional performance are sufficiently preserved such that a diagnosis of AD cannot be made by the site physician at the time of the screening visit. Additionally, all subjects were given the Alzheimer ’s disease Assessment Scale cognitive subscale (ADAS-cog), Rey auditory verbal learning test (RAVLT), Montreal Cognitive Assessment (MoCA), and the Functional Activities Questionnaire (FAQ). All study subjects met the ADNI inclusion and exclusion criteria, which have been described previously [33] and can be found at http://www.adni-info.org/. Appropriate institutional review board approval was obtained at each ADNI site and informed consent was obtained from each subject or authorized representative.

### 2.2. Image Preprocessing

A detailed description of the resting-state fMRI and volumetric image acquisition protocols can be found at http://www.adni-info.org. The fMRI and volumetric data were processed with the Data Processing Assistant for Resting-State fMRI advanced edition (http://rfmri.org/DPARSFA) [34], which is based on statistical parametric mapping (http://www.fil.ion.ucl.ac.uk/spm) and the toolbox for data processing and analysis of brain imaging (DPABI, http://rfmri.org/DPABI) [35]. Slice timing, head motion correction, and spatial normalization to the standard Montreal Neurological Institute (MNI) Echo Planar Imaging (EPI) template with a resolution of 3 × 3 × 3 mm^3^ were performed. As recommended by Anderson et al. (2011) and Murphy et al. (2009), the global signal regression was not performed to avoid introducing distortions into the time-series data [36,37]. All subjects in our study had a head movement of less than 3 mm translation and less than 3° angular rotation in any direction (out of six) during fMRI scanning. Finally, fMRI data was detrended and band-pass filtered (0.01–0.08 Hz) before conducting the functional connectivity (FC) analysis. We also performed amplitude of low-frequency fluctuations (ALFF) and regional homogeneity (ReHo) analyses, using DPABI, to investigate the intensity and homogeneity of spontaneous brain activity in the primary olfactory cortex (POC) and hippocampus.

### 2.3. Olfactory Network

Based on published fMRI task activation studies, the olfactory network includes the POC, insula, and orbital frontal cortex (OFC) [27,38]. Seed time courses were extracted from preprocessed data in MNI space ((*x y z*) coordinates) as the average time course within a five-voxel radius centered on coordinates defined from previous activation studies (see Appendix A). Seeds comprising the core ON were obtained from a meta-analysis that identified these three bilateral brain regions as most likely to be activated by olfactory stimulation [38]. They included the piriform cortex ([−22 0 −14], [22 2 −12]), the OFC ([−24 30 −10], [28 34 −12]), and the insula ([−30 18 6], [28 16 8]). Networks were computed as FC maps that survived a *p* value of less than 0.01 with a minimum cluster size (k) of 60 voxels.

### 2.4. Statistical Analysis

Demographic (age and education) and neuropsychological data (MMSE, MoCA, CDR [Clinical Dementia Rating], RAVLT, ADAS11, ADAS13, FAQ) were compared using the Kruskal–Wallis ANOVA test. Pairwise group comparisons were performed using the Mann–Whitney U test. The sex ratio between groups was compared using the chi-square test.

Whole brain and regions of interest (ROI) analyses, including group differences, were assessed using statistical inference performed at the voxel level with an FDR (False Discovery Rate) correction for multiple comparisons (*p* < 0.05) in DPABI and AFNI (https://afni.nimh.nih.gov/). Finally, an explorative correlation analysis was performed to test the significance of the relationship between FC values and the neuropsychological test scores described above.

### 2.5. Volumetric Analysis

A Bayesian model-based segmentation toolbox in the FMRIB Software Library (also known asFSL) (FIRST; http://fsl.fmrib.ox.ac.uk/fsl/fslwiki/FIRST) was used to segment each anatomical image and create vertex meshes for left and right hippocampus. Quality control of the subcortical segmentations was performed by an experienced image analyst, following FSL FIRST guidelines (https://fsl.fmrib.ox.ac.uk/fsl/fslwiki/FIRST/UserGuide). No participants were excluded because of poor segmentation of one or more structures. We computed the left and right hippocampus volumes by generating masks for both hippocampi in 3D volume space and multiplying the number of voxels in the mask by the voxel size.

### 2.6. Relationship between ON FC and Neuropsychological Test Scores

A Pearson correlation analysis was performed between the ON FC and neuropsychological test scores, including RAVLT, ADAS11, and ADAS13. The RAVLT is a list learning task that assess multiple aspects of verbal learning and memory. The ADAS11 and 13 are measures of global cognition, including memory, reasoning, language, orientation, ideational praxis, and constructional praxis. The test is scored in terms of errors, with higher scores reflecting poorer performances. Because this analysis was exploratory in nature, a statistical significance level of *p* < 0.05 was used.

## 3. Results

### 3.1. Demographic and Neuropsychological Data

Demographic and neuropsychological test score data are tabulated in Table 1. No significant differences in age, gender, and education level were found between groups. Significant differences were found between groups on neurospychological test scores (*p* < 0.001). Pairwise comparisons are tabulated in Table 2. As expected, neuropsychological test scores were significantly impaired in AD compared to the CN, EMCI, and LMCI groups.

### 3.2. ON FC Differences and the Region of Interest (ROI) Analysis

We conducted a whole brain one-way ANOVA and detected ON FC group differences in the right hippocampus, with no significant differences in the left hippocampus (Figure 1). The coordinate of the peak value voxel in the right hippocampus is tabulated in Table 3. A subsequent region of interest (ROI) analysis of the ON FC is shown in Figure 2. Of note, the ON FC is significantly different between EMCI and LMCI groups.

### 3.3. Region of Interest (ROI) Analysis of the Hippocampal Volume

The hippocampal volume analysis by FSL revealed significant differences in the right hippocampal volume between the AD group and all other groups (*p* < 0.05 FWE [Family–wise Error Rate] corrected for multiple comparisons). No significant differences in hippocampal volume were observed between the EMCI and LMCI (Figure 3). A similar pattern was observed for the left hippocampal volume (see Appendix A). No correlations between the ON FC and hippocampal volume were significant in any group.

### 3.4. Correlation between ON FC and Neuropsychological Test Scores

There was a significant positive correlation between the ON FC and RAVLT-immediate (r = 0.225, *p* = 0.0083). Conversely, ON FC was negatively correlated with CDR (r = −0.178, *p* = 0.038), ADAS11 (r = −0.195, *p* = 0.018), and ADAS13 (r = −0.193, *p* = 0.019). These exploratory correlation analyses were not corrected for multiple comparisons (Figure 4).

## 4. Discussion

The pathological cascade of AD suggested by Jack et al. (2010) highlighted the early emergence of olfactory impairments in AD. In fact, the level of olfactory impairment has demonstrated the ability to distinguish between disease stages and predict progression from MCI to AD [39]. Additionally, it has been suggested that early olfactory impairment may reflect the onset of AD, amnestic mild cognitive impairment (aMCI), and the presence (or formation) of amyloid-β (Aβ) and tau pathology in cognitively normal adults [1,40]. Furthermore, neurofibrillary tangles (NFTs) in AD are known to selectively disturb specific cortical layers in the hippocampal formation [41]. In turn, this may disrupt hippocampal projections [42] effectively isolating the hippocampus from the rest of the brain. This, combined with our finding of reduced ON FC to the hippocampus, may provide a basis for explanation of olfactory deficits in AD, corroborating the disconnection hypothesis [43]. Four main findings were generated in this study: first, the ON FC to the right hippocampus, which reflects the coherence of brain activity between the ON and the hippocampus, decreased depending on the AD disease state; second, the ON FC to the hippocampus was significantly different between the EMCI and LMCI groups; third, the ON FC to the hippocampus was a more sensitive indicator of AD progression compared to hippocampal volume in the early stages; and fourth, ON FC was meaningfully related to cognitive functions, based on significant correlation with auditory verbal learning scores (RAVLT). Thus, FC measures of the ON may offer unique opportunities to investigate direct and specific effects of local neurodegeneration to network disruption and functional deficits in AD.

Degeneration of the entorhinal cortex (ERC), part of the primary olfactory cortex (POC), affects activity in the hippocampus that memory processes (including odor-related) depend on. Neuroanatomically, an impaired entorhinal cortex will disrupt projections of the hippocampus that are necessary for successful memory formation [44,45,46]. Therefore, damage to the ERC may disconnect the hippocampus from the cerebral cortex [47,48], a proposition supported by our results of reduced FC between the ON and hippocampus. However, it should be noted that our functionally-defined ROI did not include the ERC. Nevertheless, there is support to suggest that FC is a potential marker for memory decline in the early stages of AD [20,49]. We further investigated hippocampus connectivity using the measure degree of centrality (DC), which quantifies direct connections of a given voxel with the rest of the brain [50]. As shown in Appendix A of the Appendix A, we observed decreasing DC values in the right hippocampus, dependent on AD disease state. These results demonstrate potential isolation of the hippocampus from the rest of the brain in AD subjects [51].

Seed-based FC studies of the hippocampus have found widespread connectivity impairments in AD [19,20,52]. Using similar methodology, Li et al. (2002) and Greicius et al. (2004) showed reduced synchrony of low-frequency fluctuations (LFFs) and resting-state activity in the hippocampus of patients with AD. More specifically, Wang et al. (2006) noted a decrease in the right hippocampal connectivity to the medial prefrontal cortex (MPFC), ventral anterior cingulate cortex (vACC), and posterior cingulate cortex (PCC) in AD, likely indicating decreased activity of the default mode network (DMN) and contributing to episodic memory impairment. However, the left vs. right hippocampal involvement in olfaction and AD remains unclear and is an interesting topic for future research.

Furthermore, a recent olfactory fMRI paper by Karunanayaka et al. (2019) showed reduced task related ON activation and DMN suppression in mild cognitive impairment (MCI) and AD subjects compared to age-matched cognitively normal subjects. Together, these studies are consistent with the notion that decreased connectivity in the DMN and other networks are pervasive across broad brain regions in subjects with AD [53]. Combined with the current findings, a likely contributor of AD-related olfactory deficits is the disruption of ON connectivity to the hippocampus which is linked to the DMN [54]. Previously, we proposed a mesoscale brain network model that anatomically and functionally linked the DMN to the ON via the hippocampus [55]. That model, supported by the findings of the current study (Figure 5), may help differentiate patterns of olfactory deficits and their development in AD progression, leading to new studies of AD pathophysiology focusing on coupling impairments affecting network dynamics [56].

Our study is consistent with findings of decreased connectivity in brain networks in subjects across various AD disease states. While ON FC and hippocampal volume were not correlated in any group, we observed FC differences between EMCI and LMCI subjects in the absence of significant hippocampal volume differences. Interestingly, no differences in FC were observed between LMCI and AD subjects, which did have significant hippocampal volume differences. Based on these observations, our results suggest that hippocampal volume and ON FC may provide complementary information, with the latter being a more sensitive marker for AD progression, preceding volumetric loss.

We also investigated the POC volume using methods described in the Appendix A. Although Appendix A shows a decreasing trend in POC volume dependent on AD disease state, it did not reach significance. Unlike the hippocampus, the FSL software is unable to perform automatic segmentation of the POC. Thus, we used nonlinear transformations to project the POC in the standard space onto individual subject space before calculating POC volumes. This method may have introduced significant error into our POC volumetric analysis given the close proximity to air tissue boundaries in the brain. As a result, POC volume sensitivity in AD was compromised compared to hippocampal volume.

Unlike previous resting-state studies, the current study did not find increased FC between the ON and hippocampus in AD. Further, the ALFF and ReHo analyses in the POC and hippocampus did not detect any group differences. Since the ADNI data we analyzed did not include olfactory behavioral data, it is not possible to comment on compensatory connectivity in the current analysis. However, previous studies have hypothesized that increased activity and connectivity represent compensatory activity as cognition becomes impaired [57,58,59,60]. Alternatively, animal and human studies have suggested that increased activity may, in fact, reflect ongoing damage due to AD [61,62]. The latter hypothesis is congruent with our results, as demonstrated by widespread neuronal loss affecting loss of functional coherence in olfactory brain structures, leading to decreased FC between the ON and the hippocampus.

Our exploratory analysis indicated significant correlations between ON FC and neuropsychological test scores. Functional connectivity between the ON and the hippocampus was positively correlated with RAVLT, a measure of verbal memory. Postuma et al. (2011) reported a significant correlation between episodic verbal memory and olfactory impairment [63], a result that was reflected in our finding. Additionally, FC between the ON and the hippocampus was negatively correlated with ADAS scores. Given that lower scores reflect better performance, our results suggest that intact FC is related to better performance on a measure of global cognition. Lastly, intact ON FC was significantly related to lower clinician rating on the CDR, which is a measure of disease severity and categorization. However, it should be noted that these correlations must be interpreted with caution when applied to the general population.

These findings may help establish specific relationships between ON FC and pathological changes in the POC, hippocampus, and AD-related behavioral measures. Such relationships have been previously hypothesized based on postmortem studies where severe AD pathology has been found in olfactory structures. The current results provide additional in vivo evidence to support the involvement of the ON beyond the olfactory bulb and tract in AD pathology. This gives credence to the possibility that ON connectivity could serve as a possible predictor of cognitive decline in AD.

### Limitations of the Study

Although the ON FC results survived statistical correction for multiple comparisons, our findings should be considered preliminary in the absence of clinical olfactory testing (e.g., University of Pennsylvania Smell Identification Test (UPSIT). Additionally, the ADNI data set used lacks control for potential confounding factors associated with olfactory function (e.g., nasal pathology). As such, our exploratory correlation analyses need to be replicated with the inclusion of potential confounding factors. We did not find differences in ON FC with respect to APOE4 status, in any group. Future studies should focus on investigating the relevance of olfactory testing and its applicability to those with genetic risk factors for the development of AD. Other limitations include the cross sectional nature of this data set and the absence of longitudinal RS-fMRI data, olfactory data, genetic data, and neuropsychological data. These drawbacks underscore the need for follow-up studies, with a larger cohort and a more robust data collection protocol, to better elucidate ON FC as an indicator of olfactory performance and disease state in AD progression. Nevertheless, we hypothesize that the observed reduction in FC between the ON and the hippocampus in AD and MCI groups may constitute a significant risk factor for progressive decline and, ultimately, dementia. Based on the current results, this hypothesis is necessarily speculative.

The current study included subjects with motion parameters less than 3 mm translation and 3° angular rotation in the 6 directions (using the standard motion correction procedure as implemented in Statistical Parametric Mapping 12, aka SPM 12). Therefore, given the number of subjects in the current study (121) and the focus on FC, we feel confident that we have addressed subject motion adequately in the current analysis. However, a better option would have been the framewise displacement method described in Power et al. (2019) that does address the effects of transient subject movements. This method is known to be superior at detecting spurious but systematic correlations in FC networks due to subject motion [64].

As mentioned in the introduction, evidence suggests that the number of mature olfactory sensory neurons in the olfactory epithelium is reduced in AD [65]. Therefore, future AD studies should focus on investigating the relationship between the reduced number of olfactory neurons at the periphery and the systems-level FC investigated in the current study. Studies of this nature may help delineate peripheral vs. central factors contributing to AD-related olfactory impairment. Lastly, with the expansion of knowledge related to AD pathophysiology, there are multiple ways to subtype AD patients (i.e., according to different genetic backgrounds and/or clinical presentations). Our study, however, focused on AD with memory deficits, one of the first clinical subtypes described [66]. Therefore, future studies are warranted to understand the relevance of resting-state functional connectivity differences in AD subtypes other than amnestic MCI.

## 5. Summary and Conclusions

This research provided new knowledge of the neural substrates of resting-state FC between the ON and the hippocampus and its relationship to AD. Results indicated that the loss of hippocampal tissue volume lags behind disruption of resting-state FC between the ON and the hippocampus. Importantly, the ON was not preferentially affected in later stages of AD, but was significantly involved in the early stages, which makes olfaction a potential sensitive indicator for AD progression. Based on the observed progressive decline of ON FC, our results lend support to the scientific premise that olfactory dysfunction in AD precedes the development of dementia. The results provide a strong foundation for future AD studies that focus on how prominent olfactory deficits in AD are related to neurodegeneration in the ON and hippocampus. Future research should also investigate if olfactory deficits signal progressive disruptions to the ON and its connectivity to the hippocampus. Olfactory deficits in prodromal AD patients may create a unique opportunity for rs-fMRI to directly address the functional consequences of observed neuropathological changes. In summary, our study provided in vivo rs-fMRI data showing functional connectivity degeneration between the ON and hippocampus in MCI, which provides some of the structural basis for the olfactory deficits in these patients. Specifically, significant differences in ON FC to right hippocampus were observed between EMCI and LMCI, highlighting the ability to differentiate between disease status. Importantly, ON-based FC could be used in conjunction with volumetric measurements of the POC and hippocampus and behavioral olfactory testing (e.g., UPSIT) to increase the diagnostic sensitivity and specificity of MCI patients. This development could contribute to an inexpensive, noninvasive predictor for cognitive decline in the early stages of AD [67].

## Figures and Tables

**Figure 1 brainsci-09-00338-f001:**
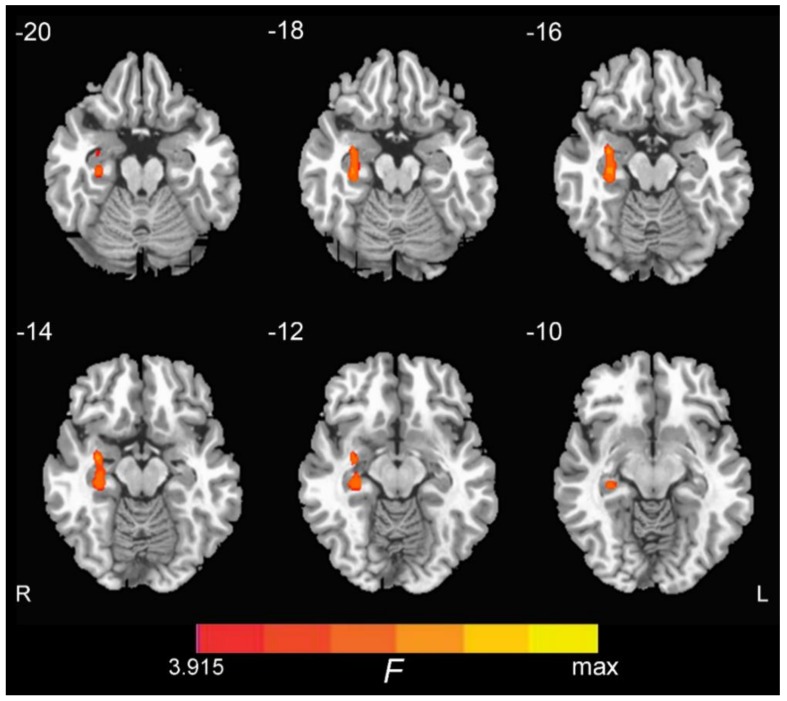
Group differences among CN (cognitively normal), EMCI (early mild cognitive impairment), LMCI (late mild cognitive impairment), and AD (Alzheimer’s disease) groups of the olfactory network (ON) functional connectivity (FC) to the right hippocampus (*p* < 0.01, AlphaSim corrected).

**Figure 2 brainsci-09-00338-f002:**
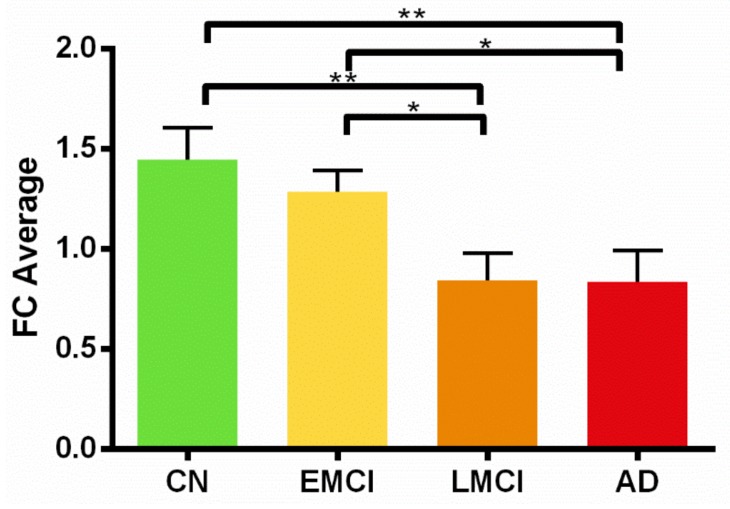
Differences in ON FC to the right hippocampus in CN, EMCI, LMCI, and AD groups. *: *p* < 0.05; **: *p* < 0.01. CN: cognitively normal; EMCI: early mild cognitive impairment; LMCI: late mild cognitive impairment; AD: Alzheimer’s disease. Of note, the ON FC to the left hippocampus was not significantly different between groups.

**Figure 3 brainsci-09-00338-f003:**
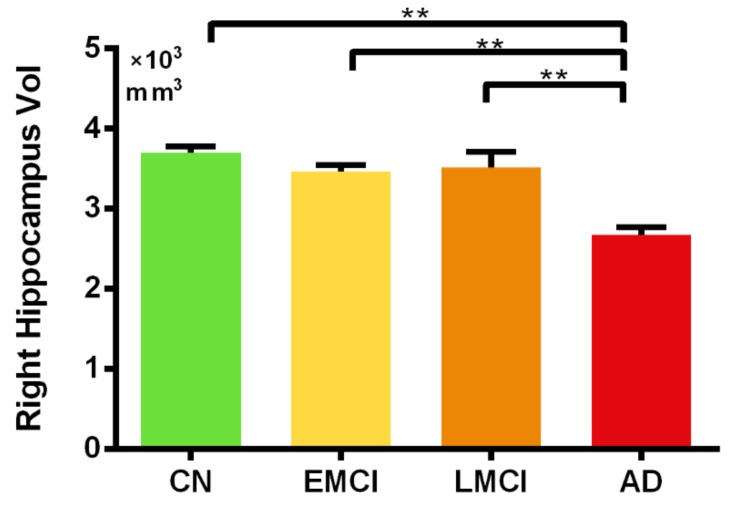
Differences in right hippocampal volume (vol) in the CN, EMCI, LMCI, and AD groups. *: *p* < 0.05; **: *p* < 0.01. CN: cognitively normal; EMCI: early mild cognitive impairment; LMCI: late mild cognitive impairment; AD: Alzheimer’s disease.

**Figure 4 brainsci-09-00338-f004:**
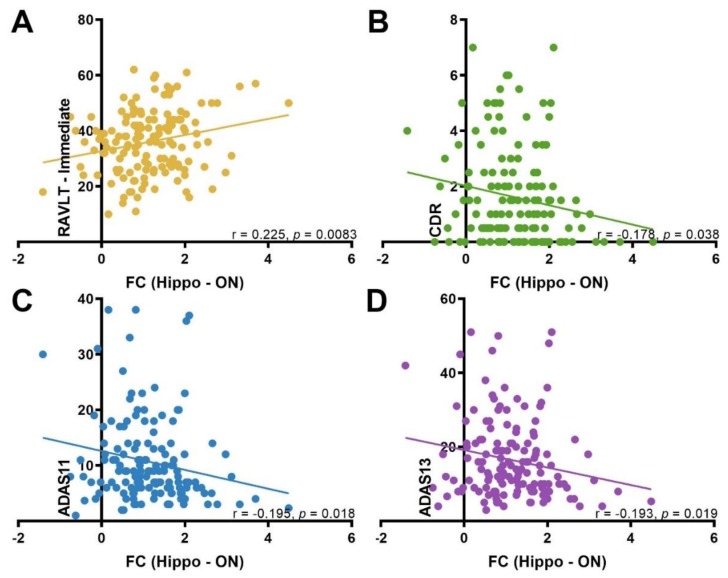
The correlations between each cognitive test score and the ON FC with the hippocampus. (**A**) RAVLT: Rey auditory verbal learning test; (**B**) CDR: clinical dementia rating; (**C** and **D**) ADAS: Alzheimer’s disease assessment scale. FC: functional connectivity.

**Figure 5 brainsci-09-00338-f005:**
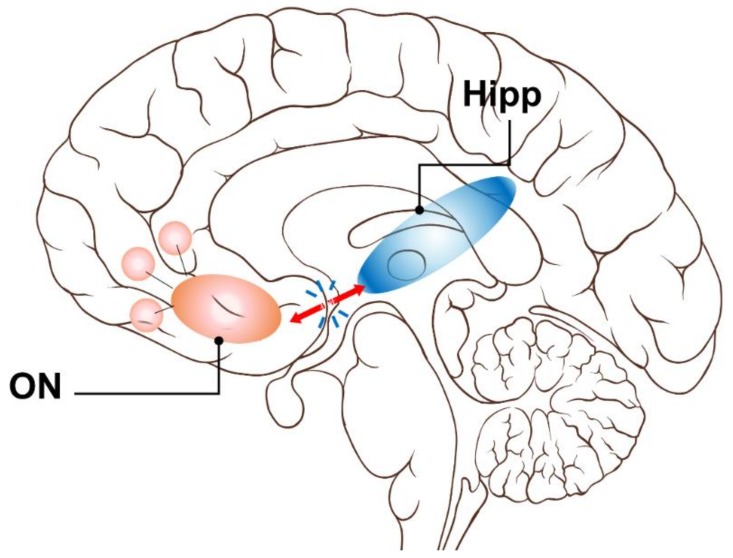
An FC model for olfactory deficits in AD. Early AD pathology compromises the ON FC to the hippocampus causing deficits in olfaction and memory. This Resting state fMRI, or rs-fMRI, model corroborates the model proposed by Lu et al. (2019) using olfactory task fMRI data. Based on the results of the current study, it is possible that the impaired functional connectivity between the ON and DMN in AD may be due to weakened or impaired ON hippocampus connectivity. Critically, this rs-fMRI model provides a testable hypothesis to relate AD neurodegeneration-to-olfactory impairment. In line with the brain network perspective, the proposed model provides pathophysiologic insight into neurodegenerative processes that may link olfaction to memory and other cognitive function deficits.

**Table 1 brainsci-09-00338-t001:** Demographics and neuropsychological data for all groups.

	CN (*n* = 44)	EMCI (*n* = 46)	LMCI (*n* = 31)	AD (*n* = 26)	*p* #
Male, No. (%)	17 (38.6)	19 (41.3)	18 (58.1)	11 (42.3)	0.518
Age	74.18 (6.1)	71.69 (7.3)	72.41 (7.4)	71.55 (7.3)	0.37
Education	16.50 (2.7)	15.65 (2.6)	16.90 (2.3)	15.31 (2.8)	0.052
MMSE	28.86 (1.4)	28.39 (1.6)	27.74 (1.6)	22.54 (2.6)	<0.001
MoCA	25.68 (2.1)	24.00 (2.8)	22.47 (3.2)	15.88 (5.7)	<0.001
CDR	0.045 (0.2)	1.42 (0.9)	1.73 (0.93)	4.46 (1.4)	<0.001
RAVLT	44.23 (8.1)	37.93 (10.3)	33.16 (7.4)	22.46 (7.7)	<0.001
ADAS11	5.69 (2.1)	7.93 (3.4)	11.09 (4.7)	23.19 (8.2)	<0.001
ADAS13	9.19 (3.7)	12.52 (5.2)	17.55 (7.0)	34.23 (9.7)	<0.001
FAQ	0.11 (0.5)	2.48(3.9)	4.871 (4.9)	15.038 (7.4)	<0.001

CN: Cognitively normal; EMCI: early mild cognitive impairment; LMCI: late mild cognitive impairment; AD: Alzheimer’s disease; RAVLT: Rey auditory verbal learning test; CDR: clinical dementia rating; ADAS: Alzheimer’s disease assessment scale; MMSE: Mini-Mental State Examination; MoCA: Montreal Cognitive Assessment; FAQ: Functional Activities Questionnaire; *: chi-square test; #: Kruskal–Wallis ANOVA test.

**Table 2 brainsci-09-00338-t002:** Pairwise comparisons between groups.

	CN vs. EMCI	CN vs. LMCI	CN vs. AD	EMCI vs. LMCI	EMCI vs. AD	LMCI vs. AD
*p*						
Education	0.095	0.632	0.048	0.046	<0.001	0.024
MMSE	0.191	0.007	<0.001	0.125	<0.001	<0.001
MoCA	0.018	<0.001	<0.001	0.053	<0.001	0.001
CDR	<0.001	<0.001	<0.001	0.344	<0.001	<0.001
RAVLT	<0.001	<0.001	<0.001	0.058	<0.001	<0.001
ADAS11	0.014	<0.001	<0.001	0.011	<0.001	<0.001
ADAS13	0.017	<0.001	<0.001	<0.001	<0.001	<0.001
FAQ	<0.001	<0.001	<0.001	0.011	<0.001	<0.001

CN: cognitively normal; EMCI: early mild cognitive impairment; LMCI: late mild cognitive impairment; AD: Alzheimer’s disease; RAVLT: Rey auditory verbal learning test; CDR: clinical dementia rating; ADAS: Alzheimer’s disease assessment scale; MMSE: Mini-Mental State Examination; MoCA: Montreal cognitive assessment; FAQ: Functional Activities Questionnaire. Pairwise comparisons were performed using the Mann–Whitney U test.

**Table 3 brainsci-09-00338-t003:** Coordinates of the peak voxels in the hippocampus.

Cluster	Cluster Size (Voxel)	MNI Coordinates			*t* Value
		*x*	*y*	*z*	
Right Hippocampus	74	33	−6	−15	7.59

MNI: Montreal Neurological Institute.

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
