# Peer review of "Functional Connectivity between the Resting-State Olfactory Network and the Hippocampus in Alzheimer’s Disease"

_brainsci, 2019, doi:10.3390/brainsci9120338_

Round 1

Reviewer 1 Report

The paper entitled "Functional Connectivity between the Resting State Olfactory Network and the Hippocampus in Alzheimer’s disease" highlights that functional connectivity between the olfactory network and the right hippocampus became progressively disrupted across Alzheimer's disease states. Moreover, a correlation between this observation andneuropsychological test scores is established. These findings lead authors to suggest that, the functional connectivity between the olfactory network and the hippocampus may be a sensitive marker for AD progression, preceding gray matter volume loss.

The observations and suggestions are really of interest in the field of Alzheimer's disease linked to "olfaction impairment " as a marker of the pathology.

Nevertheless, the manuscript should be improved:

Introduction and discussion:

Authors do not mention neither discuss about the olfactory deficits in peripheral area (in the olfactory mucosa), at least in the introduction. Olfaction impairments in AD could results both of peripheral and central deficits and this fact is not introduced neither discussed. For exemple, Zhang J et al 2018,neurobiol aging, showed that the number of mature olfactory sensory neurons in the olfactory epithelium is reduced. is this the same in the brain? Authors do not discuss the point.

Does a correlation exist between the reduced number of olfactory neurons at the peripheral level and the functional connectivity in the brain? 

Materials and methods:

The age and the sex of the subjects should be mentioned in the materials and methods sections. So the table 1 should be cited in this section of the manuscript.

The methodology used for the left hippocampus should also be reported in this section.

A section "statistical analysis" has to be added. 

Results:

Table 1: what means F?

Legends of the figures (particularly the figure 4) must be completed to explain the results. All abbreviations that appear on figures must be defined in the legends.

Typography of the manuscript:

Some mistakes have been noticed in the text:

All the Latin words have to be written in italic, in all the manuscript (et al., in vivo, via...).

line 65 : a space must be removed between task and performance.

for the statistics, p<0.05, typography needs to be verified p<0.05 or p < 0.05 with or without spaces before and after the sign. (for example, lines 167, 169 and 170, lines 183, 186).

line 176, p<.05.

line 193, a coma is missing after "In fact".

line 196, a coma is missing after "In turn". Spaces must be removed before and after "effectively isolating".

line 216: what is the signification of "(ref)?

line 239: the "M" of model in figure 5, should be written in lower case.

General aspect to improve the reading of the manuscript: too many abbreviations are used in the text, leading to an unpleasant reading. all abbreviations should be removed.

Reviewer 2 Report

The authors present with this paper an all-in-all well-conducted study on a very interesting topic. Olfactory impairment in AD (and also in other neurodegenerative disorders) is a potential biomarker and rs-fMRI is a promising, non-invasive diagnostic tool. I have, however, the following concerns about this study and they should be addressed by the authors prior to publication.

Major comments:

The paper would benefit if the authors analyzed all data in terms of framewise displacement (please refer to Power et al.: “Spurious but systematic correlations in functional connectivity MRI networks arise from subject motion”). Traditional motion regressors do not address the effects of transient subjects’ movements. The easiest approach would be to calculate framewise displacement and exclude participants that exceed the voxel size. The statistical methods used for the analysis of the demographic and neuropsychological data should be thoroughly presented in the Methods. It appears that the authors used Chi-squared test. I would advise the choice of non-parametric test, i.e. Kruskal-Wallis ANOVA with Mann-Whitney U test for pairwise comparisons. Why did the authors study FC only with the right hippocampus? Did they also analyze the left hippocampus, without getting any significant result? If this is the case, then how do they explain this unilateral impairment? The fourth main finding (“ON FC is meaningfully related to cognitive functions”) of this study is questionable. Only RAVLT appears to have a somehow positive correlation (I would recommend the use of the exact p value instead of the arbitrary p<0.01). The other graphs appear to be rather idiosyncratic and possibly not applicable to the general population. Besides, the authors declare that they deliberately chose a high significance level (p<0.05) because this analysis is exploratory. I would recommend to tone down the validity of the correlation analysis.

Minor comments:

In the second paragraph of the discussion the authors discuss the role of the entorhinal cortex in the development of AD. However, the entorhinal cortex was not used as an ROI. Why? Figure 5 does not include the DMN, even though the authors “propose a mesoscale brain network model shown in Figure 5 that anatomically links the DMN to the ON via the hippocampus”. Could you please elaborate? I am afraid your argument is not clear.

Round 2

Reviewer 2 Report

The authors have addressed the majority of my previous comments, however in a somewhat unsystematic way. There are still some adjustments that need to be made:

1: There is still no mention of pairwise comparisons. Kruskal Wallis Anova showed significant differences among groups for the neuropsychological tests, but we still do not know which groups differ significantly from one another. Pairwise comparisons (with i.e. Mann Whitney U test) in such context is common practice. Results from pairwise comparisons should be mentioned in the the first section of the results.

2: Table 1: what does the p value in the 7th column refer to? I suspect the authors report parametric and non-parametric ANOVA results. Do you expect your data to be normally distributed or not? I would assume they are not normally distributed and in that case you should only perform non-parametric tests. 

3: It should be clearly stated in both the methods and the results that you analysed both hippocampi and did not find any significant differences in the left one.

4: Regarding the significance of the correlation analysis, you have indeed added  one sentence towards the end of the discussion. However, in the first paragraph it is still stated: "fourth, ON FC is meaningfully related to cognitive functions, based on significant correlation with several cognitive test scores.", which I believe is an overstatement, given that it is significantly correlated only with RAVLT. 

5: Honestly, I am not 100% convinced about your choice not to address framewise displacement. The fact that your study includes many subjects is not a convincing argument. As counterargument one could claim that most of the subjects had significant transient movement during the scans. You have at least mentioned this as a limitation, but you should be aware that the lack of this preprocessing step limits the validity of your results.
